ecology/computational biology/statistical physics

Barro colorado, neutral model, theoretical ecology

**Author for correspondence:**
P. Villegas
e-mail: pvillegas@ugr.es

# Joint assessment of density correlations and fluctuations for analysing spatial tree patterns

P. Villegas[1], A. Cavagna[1,2], M. Cencini[1], H. Fort[3] and T. S. Grigera[1,4,5,6]

[1]Istituto dei Sistemi Complessi, Consiglio Nazionale delle Ricerche, via dei Taurini 19 00185 Rome, Italy
[2]Dipartimento di Fisica, Università Sapienza, 00185 Rome, Italy
[3]Institute of Physics, Faculty of Science, Universidad de la República, Iguá 4225, Montevideo 11400, Uruguay
[4]Instituto de Física de Líquidos y Sistemas Biológicos—CONICET and Universidad Nacional de La Plata, La Plata, Argentina
[5]CCT CONICET La Plata, Consejo Nacional de Investigaciones Científicas y Técnicas, Argentina
[6]Departamento de Física, Facultad de Ciencias Exactas, Universidad Nacional de La Plata, La Plata, Argentina

PV, 0000-0002-1627-3754; MC, 0000-0001-7073-5000; HF, 0000-0001-7899-7507

Inferring the processes underlying the emergence of observed patterns is a key challenge in theoretical ecology. Much effort has been made in the past decades to collect extensive and detailed information about the spatial distribution of tropical rainforests, as demonstrated, e.g. in the 50 ha tropical forest plot on Barro Colorado Island, Panama. These kinds of plots have been crucial to shed light on diverse qualitative features, emerging both at the single-species or the community level, like the spatial aggregation or clustering at short scales. Here, we build on the progress made in the study of the density correlation functions applied to biological systems, focusing on the importance of accurately defining the borders of the set of trees, and removing the induced biases. We also pinpoint the importance of combining the study of correlations with the scale dependence of fluctuations in density, which are linked to the well-known empirical Taylor's power law. Density correlations and fluctuations, in conjunction, provide a unique opportunity to interpret the behaviours and, possibly, to allow comparisons between data and models. We also study such quantities in models of spatial patterns and, in particular, we find that a spatially explicit neutral model generates patterns with many qualitative features in common with the empirical ones.

# 1. Introduction

Ecosystems are shaped by processes—ecological forces, e.g. seed dispersal, interactions among species of the same or different trophic level, and abiotic factors, e.g. the climate, fires, etc.—occurring on different space–time scales and different levels of organizational complexity [1]. Typically, we can have access to the processes only through the patterns they generate [2]. Thus, a key challenge of theoretical ecology is to infer the underlying processes from the observed patterns [1].

Paradigmatic examples of emerging patterns in ecology are tropical rainforests. In such biodiversity hotspots, thousands of plants belonging to hundreds of species coexist in relatively small areas [3], generating complex spatial patterns of vegetation. Such patterns can be studied at the macro-level: looking at how the number of species or the species abundance distribution change with the sampled area [2]: or, on a more detailed level, by studying the spatial distributions of trees [4–6].

Ideas from statistical physics, which studies the emergence of macroscopic properties of a system from its microscopic rules, proved to be very fruitful to understand biological systems of high-level organizational complexity. An overarching concept to understand the emergent properties of such systems is that of correlation. The study of correlations has been key to understand, e.g. the rules at the basis of collective motions in bird flocks [7], the neurons of vertebrate retina [8] or the spatial yield response in a pistachio orchard [9].

In this work, we are interested in the spatial density correlations of tree patterns, using the so-called pair correlation (or radial distribution) function, $g(r)$, which quantifies the average density of trees at distance $r$ from any individual tree, normalized by the expected value based on the mean density of vegetation [6,10]. In the ecological literature, the same quantity is also known as neighbourhood density function [4,11,12], and it has often been used to detect, e.g. clumping of trees [11] which reflects in $g(r) > 1$ values. Here, we focus on two (mainly methodological) aspects.

The first concerns how to properly take into account the biases induced by the borders of the set of points [13]. This issue (often overlooked) involves two distinct problems: knowing the borders how to reduce the biases and, more subtle, how to properly identify the (not necessarily convex) borders of a set of points. Both issues can be approached with different methods [4,12–16]. Motivated by their success in coping with bird flocks [17], we use the Hanisch method [14] to cure the biases and the $\alpha$-shapes method [15,16] to identify the borders. The latter consists of a geometric algorithm to carve out concavities from a set of points using discs of a predefined radius, and does not appear to be widely known in the ecological literature, with a few exceptions [18].

The second aspect is that it can be difficult to interpret the results of studying only behaviour of the (properly computed) pair correlation function (PCF) in isolation of other relevant quantities. This was emphasized in a recent survey of the ecological literature [5], which found that, in the face of a growing number of works on spatial point pattern analysis, and of methodological reviews on the subject [4,6,12,19,20], a large percentage of the examined studies focused only on a single observable—mainly the PCF (sometimes even neglecting border issues), or a related function. Here, we propose to study, in combination with the PCF, also the way spatial density fluctuations decay with the observation scale, as it provides useful information, especially on the large scales. In this work, we show that such decay is simply related to one of the most well-known (empirical) laws in ecology, namely Taylor's power law [21] that, as far as we know, was not put in combination with the density correlation before. This law states that the standard deviation (of time or space fluctuations) of the population size scales as a power $\gamma$ of the mean population. As reviewed in [22], such relation between fluctuations and mean is found in a wide range of disciplines with $\gamma$ typically in the interval [1/2, 1]. The value $\gamma = 1/2$ characterizes the behaviour of a homogeneous random process and of cases in which the central limit theorem applies. Larger values, $\gamma > 1/2$, typically signal the presence of non-trivial correlations or the effect of heterogeneities [22]. In particular, anomalous values of $\gamma$ have been reported for almost all the species in Barro Colorado in [23], where a closely related quantity—namely the Fano factor—was investigated. Here in addition to showing the anomaly of the exponent $\gamma$, as border bias can alter its value, we also investigate the use of $\alpha$-shapes method to mitigate such bias.

To illustrate the importance of accounting for the borders and how the combination of density correlations and spatial fluctuations can aid in the process of interpretation and, possibly, of model selection, we study the emergent spatial patterns of the Barro Colorado Island (BCI) 50 ha (1000 × 500 m²) plot. Such database comprises eight censuses (every 5 years from 1980s) of more than $4 \times 10^5$ trees and shrubs with diameter at breast height larger than 0.01 m, belonging to about 300 species, providing position and species for each plant [24]. Data and related information can be found in [25].

Aiming at a qualitative comparison with BCI data and to further exemplify the ideas here developed, we also study density correlation and spatial fluctuations in three reference models. The first one is a simple heterogeneous Poisson process, which is expedient to illustrate how inhomogeneities can give rise to large density fluctuations and misleading behaviours of the density correlation. Secondly, we study the Thomas process [26], one of the simplest instances of Poisson cluster processes [6,13], which incorporates the idea of offspring dispersed by parent trees. This model and its variants have been quite successful in fitting data [27]. But, the underlying statistical properties are a prerequisite to generate the patterns, thus putting special emphasis on the inference strategy to determine the model parameters. However, extrapolating ecological processes from these procedures is a delicate issue, as combination of different mechanisms can produce similar patterns, as highlighted in [28].

Finally, we consider patterns generated by a spatial individual-based model for a community of coexisting species, where the statistical properties emerge from the incorporated processes. There are two alternatives for such class of models, reflecting two views on how biodiversity is maintained. On one side, niche theory [29] holds species differences (in resource exploitation, reproduction strategies, etc.) responsible for their coexistence. On the opposite side, the neutral theory [30,31] assumes species of the same trophic level as equivalent and sees biodiversity as a non-equilibrium stationary state realized thanks to species influx (speciation) and random drifting toward extinction (outflux) by competition and demographic stochasticity. Remarkably, both theories describe well some macro-ecological patterns of biodiversity [31–33] and some consensus is emerging that both mechanisms are at play [34]. Spatially explicit models based on niche theory are typically defined through many parameters [33], conversely, owing to species equivalence, neutral ones need very few [35,36]. Without any claim of neutrality for real data, we opted for the latter just for the sake of simplicity. Surprisingly, this simple model gives rise to a variety of behaviours, qualitatively similar to those observed in real data, demonstrating the richness induced by simple mechanisms even in a neutral context.

The BCI plot has been much studied in the past. Some studies suggested multifractal properties of the low-canopy gaps [37], simulated through cellular automata models [38] and also compatible with neutrality conditions equipped with long-range dispersal [39]. Relying on the study of the $g(r)$, double-cluster process has been proposed to explain the spatial distribution of some selected species [19]. The PCF and other statistical quantities (such as the nearest neighbour distribution) have been also studied in [40]. Finally, a few studies have pinpointed the relevance of neutral competition to generate non-trivial spatial patterns at the single-species level [41–43]. However, to the best of our knowledge, systematic studies of the possible effect of borders on the PCF were not thoroughly conducted previously, nor in combination with the scale dependence of spatial density fluctuations (which amounts to study Taylor's Law, as discussed in the following).

The material is organized as follows. In §2, we discuss the computation of PCF focusing on some species in BCI plot and on the role of boundaries. Spatial density fluctuations and Taylor's Law are discussed in §3. Correlations and fluctuations in the above-mentioned spatial models are discussed and qualitatively compared with data in §4. Section 5 is devoted to conclusion. Technical material and some supporting results can be found in the electronic supplementary material.

# 2. Density correlation for the community and for single species

## 2.1. The pair correlation function

We start defining our main observable, namely the pair correlation (or radial distribution) function, $g(r)$, which is here used to probe density correlations in the spatial distribution of trees in the BCI plot. The $g(r)$ is proportional to the probability to have a tree at distance $r$ from any given tree and describes how trees are distributed in space at varying length-scale. In particular, given $N$ points (trees, particles, etc.) in an area $A$, the function $g(r)$ is the number of points $\mathrm{d}N(r)$ in the annular area between $r$ and $r + \mathrm{d}r$ centred on one point, averaged over all points and normalized with the expected number of neighbours for a completely random (i.e. a homogeneous Poisson) distribution with mean density $\rho_0 = N/A$. In formulae, the PCF reads

$$g(r) = \frac{\mathrm{d}N(r)}{2\pi\rho_0 r\,\mathrm{d}r} = \frac{1}{N\rho_0 2\pi r}\sum_{i,j}^{N}\delta(r - r_{ij}), \tag{2.1}$$

where $i$ and $j$ denote two points in the set of interest. For a Poissonian random distribution of points one has $g(r) = 1$, by definition. Conversely, values above 1 denote clumping, i.e. tree clustering, as generically found at

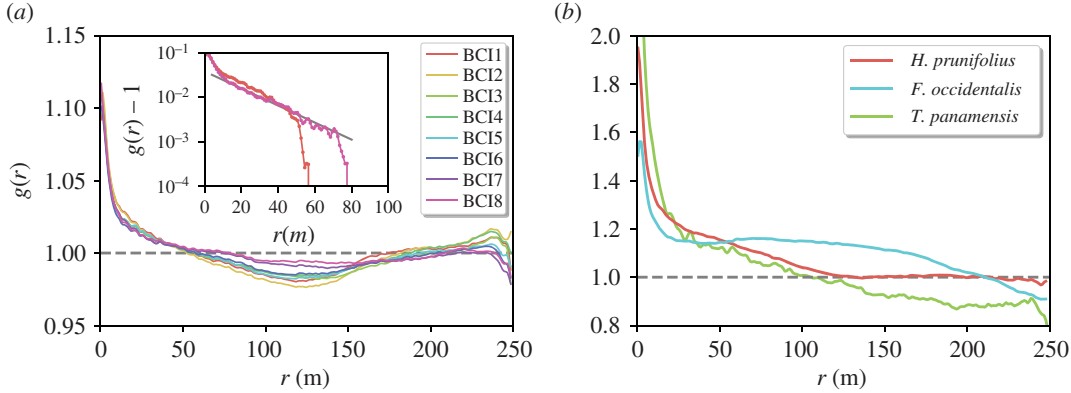

**Figure 1.** Density correlations in BCI. (*a*) Pair correlation function *g*(*r*) versus distance, for the whole tree community and different censuses. Inset: *g*(*r*) − 1 in semi-log scale for the first and last censuses; an exponential function with correlation length $\xi = 22$ m fits the data from 10 to 70 m. (*b*) *g*(*r*) for three species selected among the 10 most abundant: *H. prunifolius* (red), *F. occidentalis* (cyan) and *T. panamensis* (green). In both panels, the grey dashed line corresponds to the result for a completely homogeneous distribution.

short distances in rainforests [11], while values below 1 signal the presence of 'repulsion' or anticorrelations. In general, *g*(*r*) can have a non-trivial spatial dependence, e.g. it has spikes on regular lattices/crystal or broad peaks in a liquid [10], providing information on possible processes acting at different scales.

When computing the *g*(*r*), to avoid spurious results one has to properly take into account the presence of borders. Indeed, points close to the borders, having less neighbours than those in bulk, can bias the statistics. To mitigate or remove such biases different methods can be used. For instance, one can give different weights to bulk or border points, limit the analysis to points belonging to a subregion in the bulk or impose periodic boundary conditions by replicating the study area [4]. Here, we employ an unbiased estimator of the number of neighbours [13] originally proposed by Hanisch [14]. For each point *i*, we count any other points *j* as neighbour only if its distance from *i*, $r_{ij}$, is less than that of *i* from its closest border, $d_i$, thus constraining the sum over trees in equation (2.1). When using binning to group distances together, the point *j* is excluded if it falls on a bin that is not completely contained within the borders.

In the following, we discuss the PCF in the BCI plot both at the community level (i.e. considering all trees together regardless their species) and, for some selected species, at the single species level. We always use the Hanisch method to avoid border bias. However, this requires knowledge of the borders, a non-trivial problem with real data [17]. Thus, we compare two different definitions of the borders: the edges of the rectangular plot (which for most species approximate to the convex hull of the set), and the borders obtained with the α-shapes method [15,16]. As discussed below and detailed in the electronic supplementary material, S1, the advantage of α-shapes is to provide a geometrical criterion to remove concavities (for other methods, see [12]).

## 2.2. Rectangular borders

In figure 1*a*, we show the PCF considering all trees, regardless of their species, and for all censuses. At small distances, $r < 50$ m, *g*(*r*) > 1 provides evidence of clumping. At larger distances, the *g*(*r*) displays a plateau to a value fairly close to 1. While deviations from 1 at short distances are small, they are robust, as evidenced by removing the plateau value, where a clear exponential decay with characteristic correlation length $\xi \approx 22$ m emerges (see inset). Below 10 m, deviations suggest a steeper exponential decay though, owing to the short range of scales, precise statements are difficult. Finally, no significant differences between various censuses can be observed.

We now turn to the density correlations of conspecific trees. In particular we focus on three of the more abundant species: *Hybantus prunifolius*, *Faramea occidentalis* and *Tetragrastris panamensis*. Typically (see also electronic supplementary material, figure S6a), the PCF exhibits a common behaviour at short distances indicating clumping (*g*(*r*) > 1). Actually, the deviations from 1 at short distances are stronger than for the community-level PCF, suggesting that conspecific trees tends to be more clustered than the whole community. However, at large distances we found unexpected and diverse results. For *H. prunifolius*, the PCF converges to a plateau at 1, meaning that at large scales the

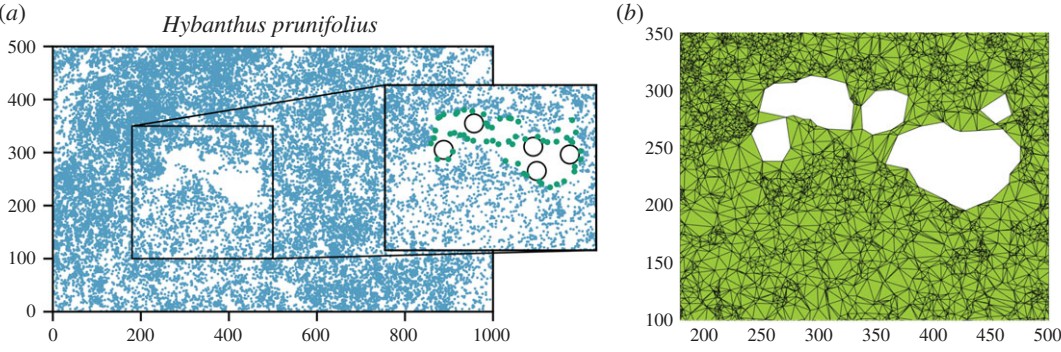

**Figure 2.** Example of non-trivial borders in the case of *H. prunifolius*. (*a*) The distribution of *H. prunifolius* individuals (points) in the 8th BCI census clearly displays a big empty region. The inset shows how using $\alpha$-shapes (a few circles with radius $\alpha = 14$ are shown) one can identify the internal border (green points represent the border trees). (*b*) Once the border trees are identified, Delaunay triangulation allows to measure the covered area, and thus estimate the average tree density, $\rho_\alpha$, with the empty region excluded.

neighbour density recovers the mean density. For the other two species, the plateau (if any) is less well defined, showing values either below 1 (*T. panamensis*) or above 1 (*F. occidentalis*). For other species (see electronic supplementary material, Figure S6a) also non-monotonic behaviours can be observed with clear signatures of anticorrelation, i.e. $g(r) < 1$.

So far we considered as borders the edges of the rectangular plot, but this is not always the most reasonable thing to do. The particular case of *H. prunifolius* offers a clear-cut example. The unambiguous empty area in figure 2 corresponds to a swampland where *H. prunifolius* cannot easily establish [44]. Clearly, when computing the $g(r)$, its perimeter must be considered as an internal border. An improper identification of the borders might bias not only the number of neighbours but also the estimation of the covered area and thus of the mean density, leading to a wrong normalization of $g(r)$.

For instance, as exemplified in electronic supplementary material, figure S2a, neglecting voids in an otherwise random distribution of points leads to spurious values of $g(r)$ above 1 (erroneously suggesting clumping), whereas the expected behaviour $g(r) = 1$ is recovered when borders are correctly taken into consideration.

## 2.3. $\alpha$-shape borders

Finding the borders of a set of points thus becomes a crucial issue. A first approximation to identify non-trivial borders is to consider the convex hull (CH) of the set of points, i.e. the smallest convex polygon containing all the points. Nonetheless, both concavities and internal empty regions are key to correctly identify the borders. Here we use the $\alpha$-shapes method [15,16], an algorithm which carves the distribution of points under consideration with a disc of radius $\alpha$, where $\alpha$ is a tuning parameter. The border is formed by pairs of trees (points) that can be touched by an empty disc of radius $\alpha$, independently of the distance between points. In other words, when a disc touches two points in the plane, these are added to the border if no other point is contained in the disc (see figure 3 for an illustration and for more technical aspects see electronic supplementary material, S1). In this way, all the concavities or voids larger than the radius $\alpha$ can be detected, identifying the set of points belonging to the non-convex border at scale $\alpha$. Clearly, for $\alpha$ large enough, the algorithm recovers the CH envelope of the system. In figure 2*a*, we show an application of the above described method to the distribution of *H. prunifolius*, one can appreciate how the $\alpha$-shape method is able to identify the internal borders. Once border trees are identified, the covered area is measured by means of Delaunay triangulation (shown in figure 2*b* for *H. prunifolius*) [45].

The case of *H. prunifolius* is particularly straightforward, as the internal borders are easy to detect by eye and, moreover, it can clearly be ascribed to soil characteristics, here a swampland. For other species, the situation is more ambiguous, and one has to bear in mind that there is no rigorous prescription to fix the value of $\alpha$, so that the method involves some level of subjectivity. In the absence of clear clues on the actual size of concavities, it is not obvious how to choose $\alpha$. In the synthetic case discussed in electronic supplementary material, figure S2, the proper value can be identified by searching for a plateau of the mean density (or enclosed area) as a function of $\alpha$, but for the BCI data such plateaus are not well

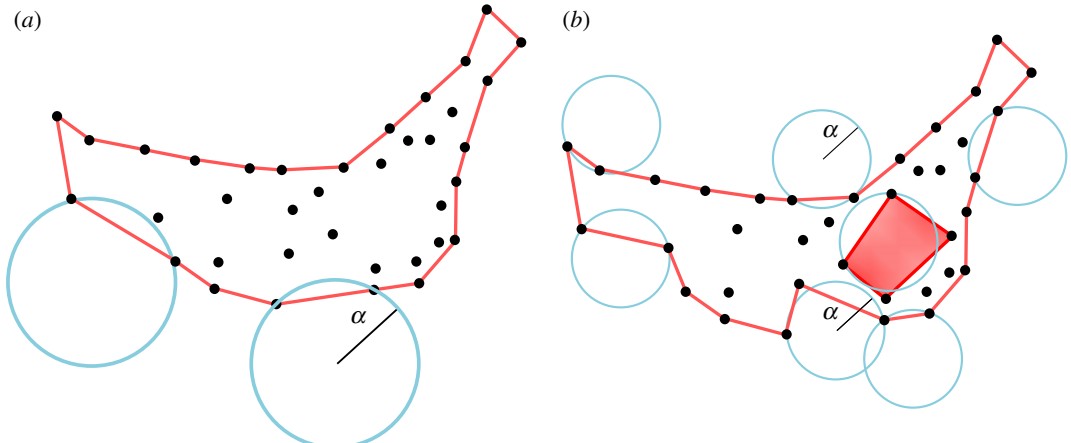

**Figure 3.** Sketch illustrating the two-dimensional $\alpha$-shape method for border identification. Given the set of points and a circle of radius $\alpha$, the circle is moved in the set of points. When the circle touches two points they are considered border points if there are no other points inside the circle. When the circle's radius is progressively reduced the algorithm is able to discriminate (a) external borders or (b) internal borders (red shaded area is now excluded) and concavities.

defined or are absent (see electronic supplementary material, figure S3). In selecting the value of $\alpha$, we required $\alpha$ to be significantly larger than the mean distance between neighbouring trees (to avoid artificial fragmentation) but small enough to ensure the identification of empty areas in the spatial distribution of different species. To check for the latter, we looked for changes in the curvature of the covered area and used visual inspection (see discussion of electronic supplementary material, figure S3). With the above proviso, we recomputed the PCFs for the three species with the borders identified with the $\alpha$-shapes. For the community level PCF, the borders essentially coincide with the rectangle. Borders for the 10 most abundant species, including the three species here discussed, are shown in electronic supplementary material, figure S4.

In figure 4a, we compare the PCF for *H. prunifolius* computed using both the naive rectangular border and those obtained with the $\alpha$-shapes, which exclude the void shown in figure 2. In this particular case, $g(r)$ exhibits a robust behaviour, being mostly independent of the border definition. Similarly, also for *F. occidentalis* (figure 4b) the dependence of PCF on the border choice is (if any) very weak. In particular, there remains a plateau at a value different from 1, meaning that the neighbour density does not recover the mean density at large scales, independently of the border definition. In electronic supplementary material, figure S6 we show the PCFs of the 10 most abundant species computed with the rectangular border and that given by the $\alpha$-shapes, respectively. Small scale clumping appears as a robust feature independently of the border, thus confirming previous findings [11].

By contrast, for some species the different choice of borders has an evident effect at intermediate and large scales. In particular, exclusion of empty areas through use of $\alpha$-shape leads to the disappearance of anticorrelations ($g(r) < 1$). However, removal of voids is a delicate issue. Indeed, such empty areas might be the result of some relevant ecological process, and thus their removal could represent the introduction of an artefact, which needs to be checked against fair biological criteria or biogeographical aspects. In this respect, the case of *T. panamensis* (which displays anticorrelation at large distances, figure 4c,d) is worth attention. In figure 4c, we compare $g(r)$ computed with the naive borders and those obtained with the $\alpha$-shapes. Here, accounting for the borders leads the PCF to a fair plateau around 1 at large scales while with the naive borders it has a quasi-monotonic decay to values clearly below 1. The comparison between different censuses is revealing. In the main panel of figure 4d, we show the PCF of *T. panamensis* computed with the naive borders for census 1 and 8, showing a clear difference at small scales between the two censuses. Conversely, the small scale behaviour is basically unchanged when considering the borders given by the $\alpha$-shapes (inset). This behaviour is explained observing that *T. panamensis* has spread between census 1 and census 8 (see insets of electronic supplementary material, figure S3b–d), so that the naive border overestimates the covered area, especially in census 1. The usefulness of using $\alpha$-shapes for spreading species was previously highlighted in [18].

The choice of borders has similarly an important effect for species contracting across censuses, as shown in figure 5 for two species that experienced steep population declines. In particular, for *Piper cordulatum* one can recognize the washing out of correlations induced by the process of disappearance of the trees (figure 5a), while for *Poulsenia armata* (figure 5b) one sees that the structure of different

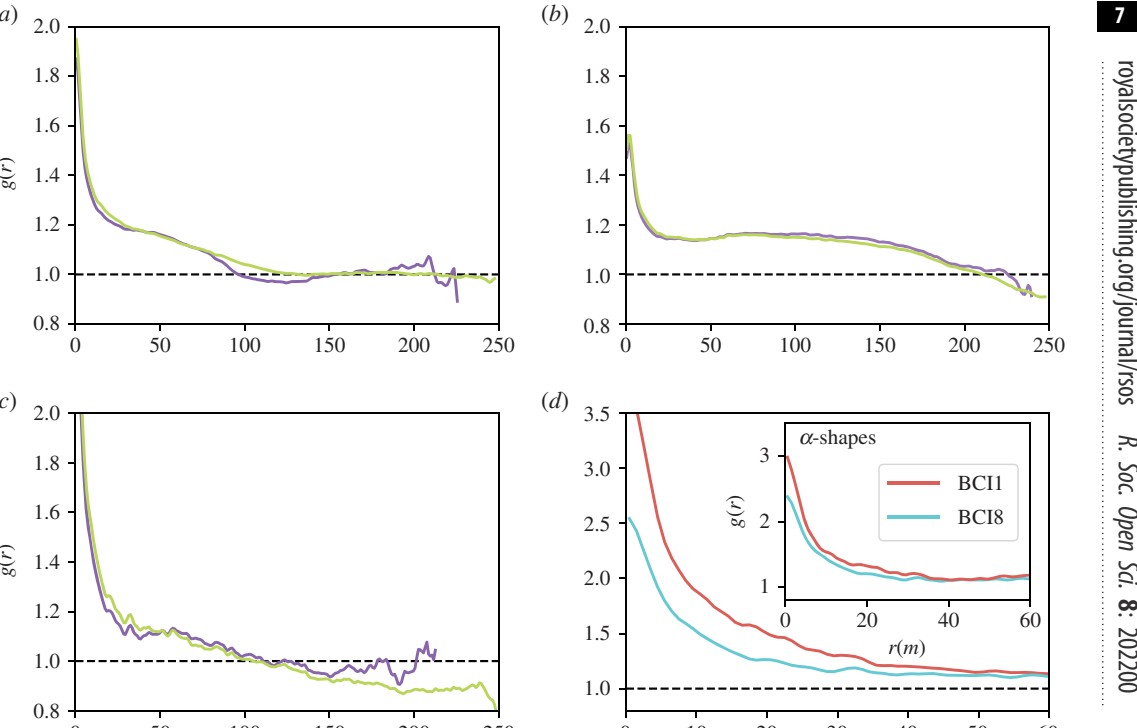

**Figure 4.** (*a–c*) Comparison of the pair correlation function, *g*(*r*), computed using the naive rectangular borders (green curves) or the borders identified via the $\alpha$-shape method (purple curves) the three selected species: (*a*) *H. prunifolius*, where $\alpha = 14$ allows to exclude the void shown in figure 2; (*b*) *F. occidentalis* for which we used $\alpha = 12$; (*c*) *T. panamensis* with $\alpha = 24$. For these and other species, the identified borders are shown in electronic supplementary material, figure S4. Panel (*d*) shows the *g*(*r*) of *T. panamensis*, computed for the first (red) and last (cyan) censuses, with the naive rectangular borders (main panel) and $\alpha$-shapes with $\alpha = 24$ (inset). In all panels, the black dashed line is the result for a uniform random distribution.

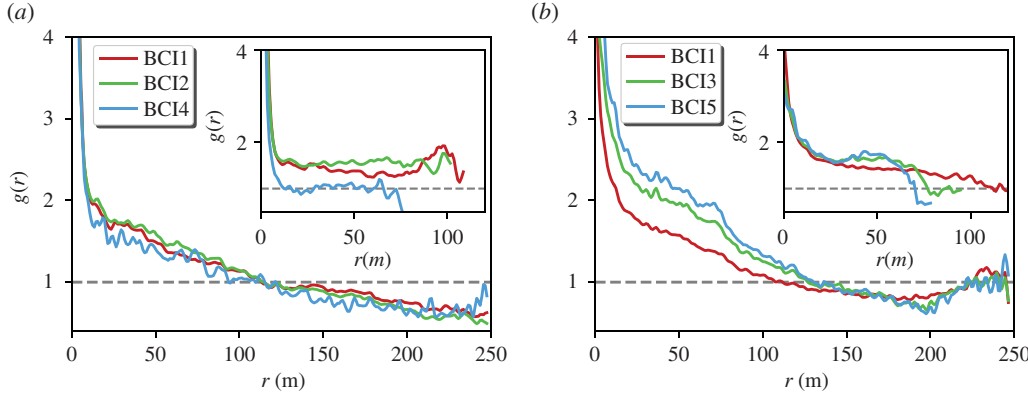

**Figure 5.** Density correlation for contracting species. (*a*) Pair correlation function for *P. cordulatum* computed with the rectangular border (main panel) and $\alpha$-shapes (inset) using $\alpha = 30$ for censuses 1 and 2, and $\alpha = 32$ for census 4. (*b*) Same as panel (*a*) for *P. armata* and censuses 1 ($\alpha = 40$), 3 ($\alpha = 35$) and 5 ($\alpha = 35$). The borders identified with the $\alpha$-shapes are shown in electronic supplementary material, figure S5.

censuses is actually quite similar, especially at small scales, contrary to what would result from the use of naive borders. Visual inspection of the tree patterns (electronic supplementary material, figure S5) also suggests that the former, while contracting, is also losing more features in the tree distribution than the latter, which qualitatively explains the difference in the density correlations.

This analysis shows that, even if the PCF—eventually with properly defined borders—displays a reasonable plateau at 1 for some cases, for others we do not recover such a trait, independently of the definition of the borders (see electronic supplementary material, figure S6). The expectation of

recovering $g(r) \approx 1$ at large scales essentially relies on two assumptions: (i) at large scales the distribution is homogeneous with a well-defined (representative) mean density; (ii) correlations have died out at the largest scales which we can observe. Clearly, the unmet expectations can originate by the breaking of one or both the assumptions. These considerations bring us to inquire about the density fluctuations, which is the subject of the following section.

# 3. Density fluctuations and Taylor's Law

For a completely random (homogeneous Poisson) process, the typical null-model in point process analysis, density fluctuations decrease with the square root of the area over which the density itself is estimated. It is instructive to see how this is achieved. Given $N$ points in an area $A$, the sample mean density is $\rho_0 = N/A$. Divide the area $A$ in cells, e.g. squares of side $r$, and denote with $n_r(x)$ the number of points in the cell centred in $x$, by definition $\rho_0 = \langle n_r(x) \rangle / r^2$ where $\langle [] \rangle$ indicates the average over all cells. To study density fluctuations, we first define the coarse-grained local density $\rho_r(x) = n_r(x)/r^2$ at scale $r$ and then look at its root mean square deviations normalized by the mean density,

$$\frac{\delta_r \rho}{\rho_0} \equiv \frac{\langle (\rho_r(x) - \rho_0)^2 \rangle^{1/2}}{\rho_0} = \frac{[\langle n_r^2(x) \rangle - \langle n_r(x) \rangle^2]^{1/2}}{\rho_0 r^2} \equiv \frac{\delta_r n}{\langle n_r \rangle}, \tag{3.1}$$

where in the last equality we used that $\langle n_r(x) \rangle = \rho_0 \, r^2$, and dropped the dependence on $x$, for simplicity. For a homogeneous Poisson process, $(\delta_r n)^2 = \langle n_r \rangle$, and equation (3.1) implies that fluctuations decay with the square root of the sampled area, $(\delta_r \rho / \rho_0) = \langle n_r \rangle^{-1/2} = \rho_0^{-1/2} r^{-1}$.

However, it is an empirical observation that for many ecological processes

$$\delta_r n \propto \langle n_r \rangle^\gamma \tag{3.2}$$

with an exponent $\gamma$ typically ranging in $1/2 \leq \gamma \leq 1$. Using equation (3.2) with $\langle n_r \rangle = \rho_0 \, r^2$ yields

$$\frac{\delta_r \rho}{\rho_0} \propto r^{2(\gamma - 1)} \tag{3.3}$$

for density fluctuations. Consequently, when $\gamma > 1/2$, and the more it approaches 1, such fluctuations become more and more important and decrease with the observation scale much slower than for a random homogeneous process.

The power law behaviour (3.2) is known in the literature as Taylor's Law (TL) and it was first put forward in the context of population ecology [21] (see Eisler *et al.* [22] for a review). The empirical relationship (3.2) is found both in spatial distribution and in the temporal evolution of biological populations, from trees to birds and insects [46,47], as well as a wide variety of systems including stock markets, heavy-ion collisions, traffic in complex networks, population ecology [48–50] and active matter, where it is known as giant density fluctuations [51,52]. Dependencies between individuals or environmental variability have been invoked to rationalize the ubiquitous emergence of such scaling law in ecology [50,53,54]. Eisler *et al.* [22] thoroughly reviews the possible mechanisms put forward to explain a deviation from $\gamma = 1/2$, which as shown above corresponds to the random (Poisson) distribution and, more generally, to cases in which the central limit theorem applies. Values of $\gamma$ in the interval (1/2, 1) are typically indicative of long correlations, a hallmark of out-of-equilibrium systems and/or the presence of spatial heterogeneities [22].

Similarly to the pair correlation function, however, the presence of borders must be properly taken into account to avoid wrong estimation of density fluctuations (electronic supplementary material, figure S2b). To cure border-induced biases, we adapted the Hanisch method [14], previously discussed for the PCF. A large number of random points is drawn in the rectangular BCI plot. Each point is retained only if it is not outside or on the borders defined by the $\alpha$-shapes, then for each valid point $i$ one measures the distance $d_i$ from its closest border and then computes the number of trees contained in circles centred in $i$ and of radius $r < d_i$. Averaging over all points (whose number should be large enough) and considering different radii, one can estimate $\langle n_r \rangle$ and $\delta_r n$. In this case $\langle n_r \rangle = \pi \rho_\alpha r^2$ as the density is estimated dividing the number of trees with the actually covered area for that value of $\alpha$ and $\pi$ accounts for using circles.[1] Apart from this inessential change, the above description of the link between TL and density fluctuations remains unchanged.

---

[1]Of course one could also use square cells, but circles are more practical in the presence of non-trivial borders.

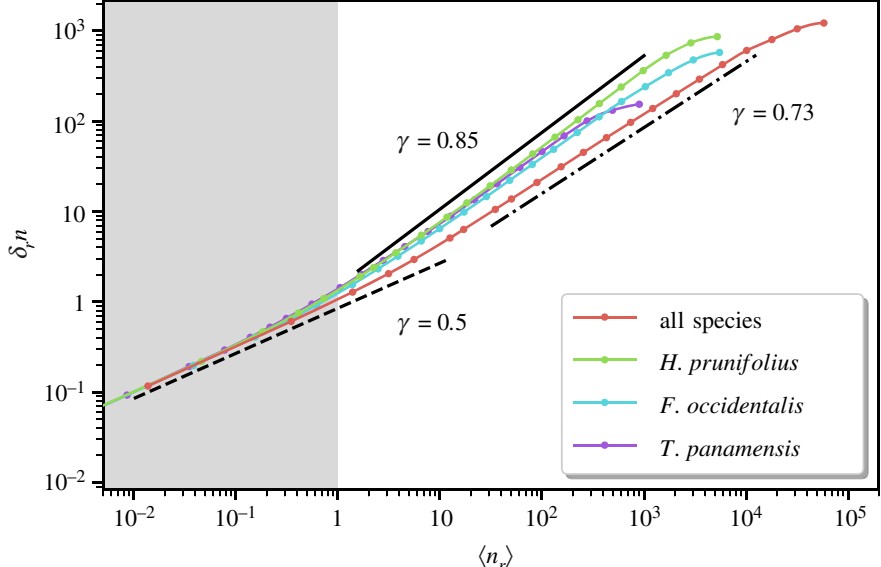

**Figure 6.** Spatial density fluctuations: standard deviation of number of trees, $\delta_r n = [\langle n_r^2 \rangle - \langle n_r \rangle^2]^{1/2}$, as a function of the mean number of trees, $\langle n_r \rangle$. Borders are identified via $\alpha$-shapes, and bias is cured as described in the text. Lines display Taylor's Law exponents $\gamma = 1/2$ (dashed, corresponding to a homogeneous random process), $\gamma \approx 0.85$ (solid, the value that describes the labelled species with some variability of the order of 0.03) and $\gamma \approx 0.73$ (dot dashed, describing the community behaviour).

Figure 6 shows that data from the BCI plot obey Taylor's Law (3.2) with exponent $\gamma$ significantly greater that $1/2$ both at the community and single species levels, except at the smallest scales (shaded area of figure 6, where $\langle n_r \rangle < 1$). However, $\gamma = 1/2$ at short scales is not due to the recovery of randomness, but to the fact that for scales much smaller than the typical distance between trees, most of the cells are empty, and a few contain a single tree, so that $\langle n_r^2 \rangle = \langle n_r \rangle \ll 1$ and trivially $\gamma = 1/2$ [22]. More in general, for very small distances, the second moment is dominated by fluctuation statistics that must be Poissonian as previously observed in [23].

At intermediate and large scales, for the previously examined *H. prunifolius*, *F. occidentalis* and *T. panamensis*, we find $\gamma$ is compatible with 0.85, while the exponent is close to 0.73 for the whole community, revealing the possible influence of heterogeneities and/or long range correlations.

Looking at the 10 most abundant species (electronic supplementary material, figure S7), we find that $\gamma$ is always greater than $1/2$, and ranges in the interval [0.75, 0.85], in fairly good agreement with previous studies [23]. The same figures shows that, for the most abundant species, the effect of borders is quite small on the TL. But the situation is different for the contracting species that we discussed in figure 5, where the change in $\gamma$ (when using properly defined borders) is quantitatively more important (see electronic supplementary material, figures S8 and S9). In particular, data for *P. cordulatum* on census 4 display the strongest difference, approaching values of $\gamma$ close to $1/2$, thus confirming the tendency toward recovering a homogeneous random distribution suggested by the density correlations.

From the above analysis, we can conclude that density fluctuations are always anomalous ($\gamma > 1/2$) for the whole BCI community and for most abundant single species. Such anomalous (with respect to the completely random process) density fluctuations observed at large scales constitute an indicator of high spatial heterogeneity and/or long-range correlations.

## 4. Theoretical considerations on modelling the data

In this section, with the aid of some illustrative reference models, we discuss the usefulness of combined information from density correlations (the PCF) and density fluctuations (Taylor's Law) when interpreting and modelling real data. In a growing order of complexity, we will consider patterns obtained with a very simple heterogeneous Poisson process (HPP), the Thomas process (TP) and a spatially explicit neutral model. These models are used only for testing how some general mechanisms influence density correlations and fluctuations, and are not proposed as explanatory models for BCI data. However, for the sake of qualitative comparison, for the HPP and TP we have chosen a rectangular domain $1000 \times 500$, similar to BCI plot, and a number of trees of the order of the

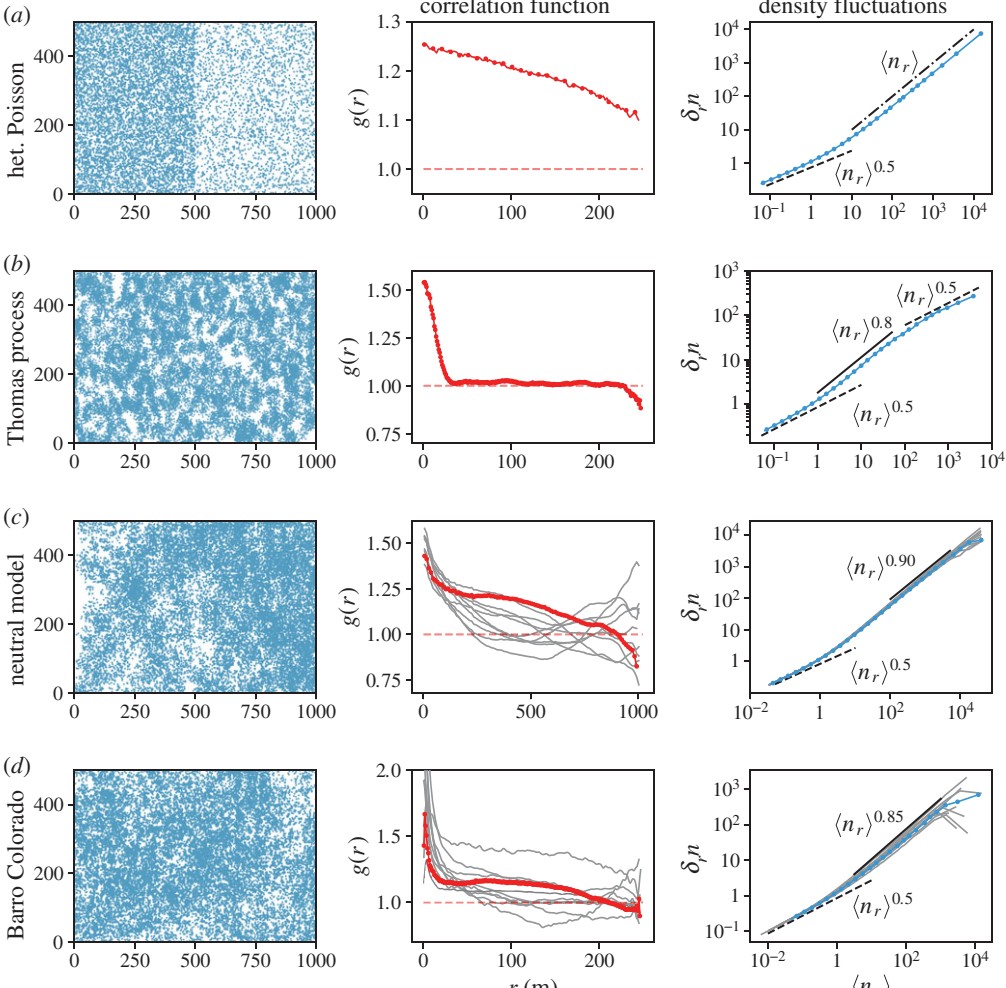

**Figure 7.** Spatial pattern (left) and the corresponding PCF (centre) and Taylor's Law (right) for the cases of (*a*) the heterogeneous Poisson process with $N = N_1 + N_2 = 3 \times 10^4$ points, $N_1 = 3N/4$ and $N_2 = N/4$ distributed on the two halves of the $1000 \times 500$ rectangle; (*b*) the Thomas process with $n_p = 10^3$ adult trees each spreading $\mu = 30$ offspring with a Gaussian kernel with $\sigma = 9$; (*c*) the multispecies voter model simulated in a $2048 \times 2048$ lattice with $\nu = 3.8 \times 10^{-6}$ and $\sigma = 9$, for which we selected 10 species among the most abundant ones (with $N \approx 1.4$–$1.6 \times 10^5$ trees) to compute the PCF and TL. For the spatial pattern, we shown only a $1000 \times 500$ rectangular portion, for an easier qualitative comparison with the other figures. (*d*) BCI data: the spatial pattern is for *Faramea occidentalis* and the PCF and TL are shown for the 10 most abundant species. In (*c*,*d*), red/blue curves refer to PCF/TL computed on the displayed pattern, while grey lines refer the rest of the 10 most abundant species. Dashed curves display theoretical expectations for a homogeneous process for PCF (red) and TL (black). The dash-dotted line in (*a*) shows $\gamma = 1$ for TL, while the solid lines shows a reference slope for the TL exponent.

most abundant species in BCI. For the neutral model, due to the characteristics of the model, a different criterion has been chosen.

## 4.1. Heterogeneous Poisson process

As discussed in §3, one of the possible origins of a Taylor exponent greater than $1/2$ (characteristic of the homogeneous completely random process) is the presence of inhomogeneities. To illustrate this point, we consider a very simple heterogeneous Poisson process, with non-uniform intensity. In particular, the $1000 \times 500$ domain is divided in two halves, each one characterized by a uniform density $\rho_1$ and $\rho_2$. In particular, we take $N = 3 \times 10^4$ points placing $N_1 = 3N/4$ and $N_2 = N/4$ of them in the first and second half of the rectangular domain, respectively.

Figure 7*a* shows a realization of the process (left) with the PCF (middle) and TL (right), computed on that instance. Notice that the $g(r) > 1$, as it is dominated by the most abundant points having density larger than the mean, which is used for normalization. A closer inspection in fact reveals that at small

distances $g(r)$ converges to the weighted average density $\rho_w = (N_1\rho_1 + N_2\rho_2)/N$ normalized by $\rho_0$. This simple example demonstrates that using $g(r) > 1$ as the sole criterion for clumping can be misleading. Here the visual inspection of the point patterns confirms that it is not clumping that leads to $g(r) > 1$.

In general, in the presence of heterogeneities, one of the main problems is the normalization with the mean density $\rho_0$, which does not represent the true density in any region of the space. This is clearly demonstrated by looking at the density fluctuations. The latter displays the trivial $\gamma = 1/2$ at very small scales, where $\langle n_r \rangle < 1$. While, at the interesting scales, it shows a clear power law behaviour with $\gamma = 1$, meaning that density fluctuations remain constant over the scales (see equation (3.3)). This looks trivial given the way the patterns has been generated; however, values of $\gamma \approx 1$ can be observed also for non-trivial reasons [22,51,52].

This example is deliberately oversimplified, in natural point process the density field is not expected to vary like a step function. Less trivial heterogeneities may lead to exponents $1/2 < \gamma < 1$, and the $g(r)$ may be more complicated.

## 4.2. Thomas process

Small-scale clumping appears to be a robust feature of rainforests [11]. As the HPP example shows, heterogeneities may lead to $g(r) > 1$ and, from an ecological point of view, this may be due to the presence of more favourable terrain, enhancing the chances of establishment and/or the number of offspring dispersed by an adult tree. However, seed dispersal limitation alone may generate clustering [11]. Here, we consider the simplest kind of point process able to mimic such clustering mechanisms, i.e. those belonging to the class of Poisson cluster processes [6,13] and, in particular the Thomas process (TP) [26]. TP assumes that: (i) $n_p$ original centres are distributed randomly according to a homogeneous Poisson process with density $\rho_p$, and (ii) each centre generates—following a Poisson distribution—$\mu$ offspring, deployed with a Gaussian kernel around the central tree with standard deviation $\sigma$, mimicking the dispersal distance of seeds.

Figure 7b shows a single realization with $n_p = 1000$, $\mu = 30$ and $\sigma = 9$ (chosen to have the small-scale density of neighbours in the range of values observed in the selected species of the BCI plot), with associated PCF and TL. The PCF displays clear signs of short-range clumping and a plateau at $g(r) = 1$ for large scales, meaning that correlations die out and the density of neighbours converges to the mean density $\rho_0$. The TP is widely employed in the ecological literature, as the PCF has a simple Gaussian-like analytical expression [6,13], $g(r) = 1 + \exp[-r^2/(4\sigma^2)]/(4\pi\sigma^2\rho_p)$, which can be used for fitting data [19,55]. From the above expression, one can readily see that the deviation from 1, and thus the intensity of clustering, is controlled by both the dispersal distance ($\sigma$) and parents density ($\rho_p$). At large scales, owing to the random distribution of the primary trees, the process recovers a homogeneous distribution and thus a well-defined mean density. This can be appreciated, even without knowing how the process was generated, by looking at the density fluctuations. At large scales indeed, the TL exponent $\gamma$ approaches the random distribution value $1/2$. At shorter scales, but above the inter-particle distance (marking the end of the trivial short-range $\gamma = 1/2$ regime), an exponent $\gamma = 0.8$, incidentally close to the typical values of BCI species, can be observed. At these scales correlations, clumping and associated inhomogeneities in the density of points are at play, leading to $\gamma > 1/2$. It is interesting to contrast the behaviour of the density correlation and fluctuations with *H. prunifolius*. There, while the $g(r)$ was reaching a plateau compatible with 1 at large scales (figure 4a), suggesting that correlations have died out, the density fluctuations (characterized by $\gamma \approx 0.85$ at all available scales) show a slower recovery of homogeneity with respect to the random process. This information could not have been obtained looking only at the PCF.

## 4.3. Spatially explicit neutral model

We now study single species patterns generated by the multispecies voter model (MVM), a spatially explicit neutral model [35,36]. Here, statistical properties are not pre-assigned as in the previous examples. They emerge from the underlying processes: dispersal limitation, demographic fluctuations and competition. Within the neutral framework, species are equivalent at the individual level: their birth/death rates and dispersal mechanism are the same and all compete for space [30]. The MVM [35] incorporates such ideas as follows. Consider a square lattice of size $N = L^2$, in which each site is always occupied by a tree. At each time step a random tree dies and is replaced: with probability $(1 - \nu)$, by a copy of a random tree in its neighbourhood (dispersal); with probability $\nu$, by a tree of a brand-new species (speciation). The neighbourhood is defined via a dispersal kernel (here a Gaussian

with standard deviation $\sigma$). Provided the dispersal length is finite and comfortably larger than the lattice spacing, the kernel functional shape is not too important [56]. Within this model species appear by speciation, grow and disappear due to demographic stochasticity and competition (controlled by local abundances) with other species, generating a (non-equilibrium) stationary state with the number of species fluctuating around a mean value, fixed by $\nu$ and $\sigma$. We numerically generated patterns at such stationary state exploiting the duality of MVM with a system of coalescing random walkers with an annihilation rate [35,57] (see electronic supplementary material, S3 and [36] for details).

Given the model characteristics, we cannot fix *a priori* the number of trees or the covered area. Moreover, the generated point patterns will depend in a non-trivial way on parameters $\nu$ and $\sigma$, whose systematic study, though interesting, is out of the scope of this work. For qualitative comparison, we required the relative rank abundance obtained with the MVM to mimic that of the BCI plot (electronic supplementary material, figure S10), and the small-scale density of neighbours of the most abundant species to be in the range of values observed in BCI data. With these two criteria, we could fix $\nu = 3.8 \times 10^{-6}$ and $\sigma = 9$, highlighting that small variations around these values do not change the results.

In figure 7c, we display a typical spatial pattern generated with the MVM. We also show the density correlations and spatial fluctuations for several species in the same class of abundance. Besides small-scale clumping is a common feature for all species, we observe diverse behaviours (with plateaus at 1 or different from 1 and also non-monotonic behaviours) at large distances, in remarkable qualitative agreement with those observed in the most abundant BCI species, shown in figure 7d. We have also checked that exclusion of empty areas through the use of $\alpha$-shape only leads to the disappearance of anticorrelations ($g(r) < 1$, see electronic supplementary material, S3.1). However, in this specific case the 'ecosystem' is homogeneous by construction and empty areas are just a dynamical effect arising from neutral competition and dispersal limitation. Thus, there is no reason—certainly not from the ecological and geographical environment—to remove such empty areas. Moreover, as shown in the right panels of figure 7c,d, the density fluctuations of the patterns generated by the neutral model display, at all available scales (above that for which $\langle n_r \rangle \approx 1$), a non-trivial scaling behaviour with an exponent $\gamma$ in the range 0.85–0.90, compatible with those observed in BCI data where $\gamma \approx (0.75 - 0.85)$.

The MVM is, by construction, spatially homogeneous, as each site of the lattice is equivalent to the others, therefore the variability observed in the PCF cannot be attributed to (extrinsic) spatial heterogeneity. On the other hand, the behaviour of the density fluctuations—which we have verified to be insensitive to the inclusion of $\alpha$-shapes method in the neutral model (see electronic supplementary material, S3.1)—points in the direction of a highly heterogeneous process as witnessed by the anomalous value of $\gamma$. This is confirmed by the visual inspection of the point process (left panel of figure 7c and electronic supplementary material, figure S11) with, possibly, persistent correlations in spite of the rather short (with respect to the system size) dispersal distance.

# 5. Conclusion

We studied spatial tree patterns by analysing the scale dependence of density correlations, probed via the pair correlation function, and of spatial density fluctuations, which are tightly linked to Taylor's power law. In particular, these tools were employed to study several species of the Barro Colorado Island plot [25]. We showed that, in order to properly estimate both observables avoiding spurious behaviours, the borders of the census plot must be treated carefully. Properly dealing with borders entails two issues. The first is avoiding the biases introduced by the points near the border, which we solved by the Hanisch method [14]. The second, and more delicate, is to identify the borders. We showed how the $\alpha$-shapes algorithm [15,16] can serve such a purpose, though with some unavoidable level of subjectivity. The $\alpha$-shapes were revealed to be particularly important when analysing expanding (as e.g. *T. panamensis*) or contracting (e.g. *P. cordulatum* or *P. armata*) species across different censuses (see also [18] for previous observations in this regard).

The small-scale behaviour of the density correlations confirmed the prevalence of tree clumping [11]. Also, we found that conspecific trees are generically more clustered than the whole community. For expanding and contracting species, clumping intensity does not seem to depend much on census (provided borders are properly identified), with the exception of *P. cordulatum*, for which density correlations and fluctuations suggest a tendency toward increased homogeneity and loss of correlations with decreasing abundance. Conversely, the large-scale behaviour of density correlations is much more complex. For some species, a plateau fairly close to the expected value of a homogeneous random process was found, while for others the plateau was at different values (non-monotonic behaviours were

also found, but they tend to disappear when $\alpha$-shape borders are implemented). These hard-to-interpret results were partially clarified by the analysis of density fluctuations, whose scale decay was shown to be related to the exponent of Taylor's Law. For most species we found that Taylor's power law exponent is larger than $1/2$ (the value of a homogeneous process), suggesting the presence of spatial heterogeneities and/or long-range correlations.

The usefulness of the joint assessment of density correlations and fluctuations was further demonstrated by analysing them in three models for point patterns, which also served for a qualitative comparison with the field data. In particular, with a very simple heterogeneous Poisson process we exemplified how inhomogeneities lead to density correlations typical of clumped distributions at mid-scales and trivial strong spatial density fluctuations. We then examined the Thomas process, belonging to the class of Poisson cluster processes which statistically mimic the dispersal of offspring by adult trees, which are randomly and homogeneously distributed. Here, the correlation function exhibits clumping at short scales and, for the chosen parameter values, spatial density fluctuations are characterized by two power laws: one with anomalous exponent at intermediate scales (due to the correlations and inhomogeneities caused by the clusters of offspring), and one with exponent $1/2$ at large scales (corresponding to the homogeneous random process controlling the distribution of adult trees). This is different from what is observed in field data. Although one could probably better mimic the empirical data with different choices of the parameters and more ad hoc clustering models—e.g. drawing the parent trees with a more complicated process, like a double-cluster process for *F. occidentalis* [19], or using different dispersal kernels—it is not obvious whether it is possible to fit simultaneously both the density correlations and fluctuations.

The above models with pre-assigned statistical features may surely serve as 'fitting models', or to illustrate a particular effect, but it is doubtful that they can be useful as 'explaining models'. In this respect, we think that the use of individual-based models, in which the statistical properties of the point patterns are not imposed but arise as a result of local dynamical rules, can be more interesting. Indeed individual-based models may present a theoretical playground to build more stringent strategies for inferring the ecological process from the generated pattern, which can then be implemented in real data analysis. This view is partially motivated by the ability of the spatially explicit neutral model we studied to produce patterns characterized by spatial density correlations and fluctuations in qualitative agreement with field data.

This programme will probably require introducing other tools and observables, besides density correlations and fluctuations. Interesting steps in this direction have been undertaken, e.g. using wavelets [28], although applied to single species. Spatially explicit neutral models or stochastic niche models [33] are able to simulate entire communities with known rules, adding effects due to competition or heterogeneities, which are surely playing a pivotal role in ecological patterns and are difficult to be inferred. In this respect, it is surprising that a simple neutral model with the (possibly unrealistic) assumption of species equivalence can not only reproduce macro-ecological biodiversity patterns [31,36] but also single species tree patterns, at least qualitatively. This suggests that the neutral model can be used as a null model against which to compare tree patterns also at the single species level.

We hope our work will stimulate the study of point patterns generated by neutral or niche spatially explicit models, which can lead to designing better observables and tools for understanding the ecological processes underlying the observed patterns.

Data accessibility. Public available data (BCI) at doi:10.15146/5xcp-0d46 [25].

Authors' contributions. A.C., T.S.G., H.F. and M.C. designed the research; P.V. performed the research, analysed the field and numerical data. All authors discussed results. P.V. and M.C. wrote the paper with input and revision from all the authors.

Competing interests. We declare we have no competing interests.

Funding. This work was supported by ERC grant RG.BIO (grant no. 785932) to A.C., and ERANet/LAC grant CRIB (project ELAC2015/T01-0593) to A.C., H.F. and T.S.G. T.S.G. also acknowledges support from CONICET, ANPCyT and UNLP (Argentina).

Acknowledgements. We thank M. A. Muñoz and S. Pigolotti for very useful comments.

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
