## [Reviewer comments · Royal Society Open Science]

Review History

Decision letter (RSOS-202200.R0)

Dear Dr Villegas:

It is a pleasure to accept your manuscript entitled "Joint assessment of density correlations and fluctuations for analysing spatial tree patterns" in its current form for publication in Royal Society Open Science. The comments of the reviewer(s) who reviewed your manuscript are included at the foot of this letter.

Please ensure that you send to the editorial office an editable version of your accepted manuscript, and individual files for each figure and table included in your manuscript. You can send these in a zip folder if more convenient. Failure to provide these files may delay the processing of your proof.

You can expect to receive a proof of your article in the near future. Please contact the editorial office (openscience_proofs@royalsociety.org) and the production office

(openscience@royalsociety.org) to let us know if you are likely to be away from e-mail contact -- if you are going to be away, please nominate a co-author (if available) to manage the proofing process, and ensure they are copied into your email to the journal.

on behalf of Dr Pietro Cicuta (Associate Editor) and Dr Pietro Cicuta (Subject Editor).

Associate Editor Dr Pietro Cicuta Comments to Author:

I think the authors responded well to the reviewers of Interface, and the paper can be published.
